# An Entropy-Based Approach to Measuring Diet Adherence

**DOI:** 10.3390/e25091258

**Published:** 2023-08-24

**Authors:** Curtis Huffman, Ana G. Ortega-Avila, Héctor Nájera

**Affiliations:** 1Programa Universitario de Estudios del Desarrollo, Antigua Unidad de Posgrado, Campus Central, Universidad Nacional Autónoma de México, Ciudad Universitaria, Mexico City 04510, Mexico; 2Instituto de Geografía, Circuito de la Investigación Científica, Universidad Nacional Autónoma de México, Ciudad Universitaria, Mexico City 04510, Mexico

**Keywords:** diet adherence, food acquisition, home food environment, food baskets

## Abstract

The aim of this study is to put forward an approach designed to calculate and sum up discrepancies between the actual food acquisition or intake and any standard or norm. Based on secondary analysis of cross-sectional data from the Mexican National Survey of Household Income and Expenditure, our proposed method to produce classes of entropy-based Diet Adherence Indices generates a Food Basket Adherence Index (FBAI) for Mexico City (2129 households). Findings suggest that it is possible to measure and decompose diet adherence using a cross entropy measure. Using food expenditure data and a normative food basket for Mexico City results, we show households’ deviations from the suggested norm for different food groups. The average FBAI was 0.44, far above the minimum score of 0 which would indicate full adherence to the normative food basket. Our measure has a distribution wide enough to detect meaningful changes and distinguish between groups with known differences, providing important new insights on the linkages between home food environments and income distribution, and food insecurity and household distribution.

## 1. Introduction

A healthy, balanced, and diverse diet is one of the foundations of a population’s well-being [1,2,3]. Current dietary patterns in Mexico indicate that 42% of the adult population follow a westernized dietary pattern, characterized mainly by high consumption of sugar-sweetened beverages, fast food, baked goods, and sweets and salty snacks [4]. The effects of unhealthy diets on major noncommunicable diseases (cardiovascular disease, type 2 diabetes mellitus, and certain types of cancer) have been well documented for these food items, and increased intake of energy, saturated fat, protein, and sugar has been associated with obesity and being overweight [5].

Diet standards are thus important for guiding and evaluating a population’s decisions as well as for the design, monitoring and evaluation of public health policies. The study of diet adherence (i.e., the extent to which a population meets such norms) is vital to keep track of divergences across different groups and populations, and it contributes to the better design of public health and social development policies [6,7,8].

Research on diet adherence demands contrasting observed data with a diet standard to draw conclusions about how far or close a population is from such a norm [9,10]. A major branch in the academic literature has aimed to address the problem of producing sound diet norms that consider the provision of energy and nutrients, cost, and cultural validity [11,12,13,14]. Given a standard, from an analytical perspective, a follow-up challenge is to find a robust procedure to calculate and compare the extent to which a population and sub-populations deviate from such guidelines. The answer to this question requires an approach that produces consistent metrics to make inter- and intraunit (units can be households, people, or countries) comparisons.

The literature that has aimed to evaluate diet adherence to specific norms lacks agreement about how to produce sound, cost-effective, and comparable metrics to examine the degree of fulfilment of diet standards [15]. The most common strategy used to derive a total score or index for evaluating adherence to prespecified guidelines or recommendations usually approaches the matter by summing points when the minimum intake cut-offs of nutrients, foods or both are met. Sometimes these indices subtract points when the components are deemed to contribute excess calories to the detriment of energy-dense foods [15]. A problem of having different composite scores is that, even when somehow validated [16], they make comparisons across samples, standards and populations extremely difficult, with no simple (structured and replicable) way to compare divergences from different cultural dietary norms. For example, several studies rate participants’ responses in such a way that the maximum and minimum values are different across and within studies and, even when normalised, do not result in meaningful comparisons of the extent of diet divergence between and within groups [17,18,19,20].

To make progress in the measurement of diet adherence, and following a long-standing tradition in inequality research in economics, in this paper we propose a novel approach based on decomposable divergence methods, i.e., indices for within and between-group comparisons [21,22]. This paper develops a robust and flexible proposal to produce different classes of decomposable entropy-based Diet Adherence Indices (DAI) in order to consistently assess a population’s divergences from either expenditure (food baskets) or diet norms by means of standard inequality measures [23]. Provided that dietary norms can be meaningfully expressed as shares across food groups, this approach helps to create comparable and decomposable (across food groups) figures of diet adherence that can be correlated with explanatory variables. It is applicable to both current food expenditure and food-intake data. A secondary objective is to present a real-data application of our proposal by evaluating the population’s adherence to a normative food basket for Mexico City. We decompose the resulting Food Basket Adherence Index (FBAI) and incorporate explanatory variables such as food security and socioeconomic factors (income, household distribution) to understand the divergences from the norm.

## 2. Data and Methods

### 2.1. Data

#### 2.1.1. Normative Food Basket

An essential condition for producing metrics on diet adherence is having diet norms or standards. The Mexican diet has a high degree of variability across regions [24,25], and any attempt to make meaningful inferences on diet adherence is conditional on the development of sensible standards for the populations of interest. We build upon a study commissioned by the Council for the Evaluation of Social Development of Mexico City (Evalúa CDMX) for the construction of a Normative Food Basket (NFB) [26,27,28]. One of the primary purposes of the study was to profile the dietary practices of the citizens of Mexico City [28]. Based on home food inventories, family food diaries, and gender-based focus groups (with mixed ages and social strata), it was possible to identify a list of popular food items and prepared dishes eaten by families at home and the factors that determine their consumption.

The final list of food items was evaluated by a group of experts and was used to create six food baskets for different age groups and genders. The food baskets account for energy and nutritional requirements, access to food, and sociocultural factors relevant to Mexico City [28]. We rely on this NFB method because it is sound in that it has the features of what are considered best practices in the region [29,30]: it can be traced back to a theory, it relies on empirical scrutiny, and, very importantly, it has face validity; a decisive, but often overlooked aspect, for an acceptable and useful standard from a policy perspective. Sadly, this is the only NFB of its kind in Mexico.

#### 2.1.2. Food Acquisition and Food Security Data

A second source of input for the implementation of our method is data on food intake or expenditure. In this example, we conduct a secondary data analysis on the Mexican National Survey of Household Income and Expenditure 2018 (ENIGH for its Spanish acronym). The ENIGH survey collects detailed information about all foods purchased or otherwise acquired (as payment in kind, gifts from other households, self-consumption, public transfers) by a household over the course of seven days. The primary respondent (the adult mostly in charge of food purchases) in each household records purchases (either for consumption or storage) in food diaries, excluding recreational food (acquisitions for special events) and gifts to other households. At the end of each day, the primary respondent undertakes an in-person interview regarding any potential mistakes made in completing the diary. The data set contains information on 247 food items, distinguishing between food-at-home (FAH) and food-away-from-home (FAFH), including weight (volume), price, total amount paid, and place of acquisition.

The ENIGH also includes an adaptation of the Latin American and Caribbean Food Security Scale (ELCSA for its Spanish acronym) [31], which is the base of the Mexican Official Poverty Measure. We include the Mexican Food Insecurity Scale, a short 12-item version of ELCSA, to correlate the FBAI with experiences of hunger, lack of variety of food and food deprivation.

#### 2.1.3. Socioeconomic Data

A definite advantage of household expenditure surveys such as ENIGH is that they usually contain detailed information about socioeconomic characteristics of the household that may influence food purchases. For the purpose of exploring the divergence from the norm in greater depth, variables such as income, household distribution, and household structure are included in the analysis as explanatory variables.

ENIGH 2018 contains a total of 74,647 households, with a relatively small (n=2129), but representative sample of Mexico City households (roughly 2.7 million households with an average size of 3.3, 1.7 SD people per household), who completed the survey between August and November 2018 [32]. Although the sample was not stratified by month, the subtropical highland climate of Mexico City (2240 m; 7350 ft) makes the effect of weather on dietary habits less of a concern.

### 2.2. Methodology

Our approach involves two steps. The first applies a norm to classify the available data on food consumption or expenditure according to food groups or specific diets (see data analysis section). The second implements an entropy-based approach to generate from the chosen norm a diet adherence index capable of decomposing the results by food or population groups. Therefore, the main outcome of our approach is a class of index with a scope that depends on the characteristics of the selected norm and the available data, i.e., the proposed approach could yield results on food quality or diet adherence indices that rely on either consumption or expenditure data.

For the applied example, we rely on a food basket that balances quality and cultural validity, and therefore the resulting index is a Food Basket Adherence Index (FBAI). We chose this for our example because it is generalisable with wide-ranging implications due to the availability of food baskets and expenditure data worldwide.

#### Entropy as a Measure of Discrepancy

In assessing a household’s diet adherence, we propose a measure that resumes the aggregate discrepancy between the empirical and the normative food group distribution. To do so, here we build upon well-known measures of distributional change [21,22], an approach that extends the generalized entropy studied in inequality measurement [23]. That is, we measure diet adherence by a population group as the divergence between its FGS weight distribution and that of the counterfactual NFB, specifically customized according to the demographic characteristics of the group in question (e.g., household structure). Consider *N* food groups and staples (FGS), FGS={1,2,…,N}, and the associated vector of non-negative weight shares $W^{e}=\left\{w_{1}^{e},w_{2}^{e},\ldots ,w_{N}^{e}\right\}$. Our approach compares the empirical distribution $W^{e}$ with a reference (normative) distribution $W^{r}=\left\{w_{1}^{r},w_{2}^{r},\ldots ,w_{N}^{r}\right\}$. Endowed with both $W^{e}$ and $W^{r}$, the quality of food acquisitions is now measured by summarizing the discrepancies between both vectors into a scalar measure.

Our estimates are presented in terms of the Kullback–Leibler divergence (KLD) [33] (the key difference between a divergence and a distance measure is that the former neither needs to be symmetric nor to satisfy the triangle inequality), a particular case of a broader class called *Bergman divergences*:(1)$$D_{\phi }^{n}\left(x||y\right):=\sum_{i\in N}\left[\phi \left(x_{i}\right)-\phi \left(y_{i}\right)-\left(x_{i}-y_{i}\right)\phi^{{}^{\prime }}\left(y_{i}\right)\right],$$
where ϕ(c):=clnc, *x* represents a given distribution, *y* an objective distribution and *N* the coarse-grained states across which units of analysis are distributed. Thus, for our estimates of food acquisition quality, we aggregate divergences between the normative and the empirical FGS weight distributions as follows:(2)$$D(W^{e}\left|\right|W^{r})=\sum_{n=1}^{N}w_{n}^{e}\ln\frac{w_{n}^{e}}{w_{n}^{r}}+w_{n}^{r}-w_{n}^{e}.$$
where the last two terms add up to zero across the sum. The KLD function is perhaps the most well-known information theoretic measure of discrepancy between distributions. It is well known that $D(W^{e}||W^{r})\geq 0$ and the equality holds if and only if $W^{e}=W^{r}$ (the properties of this measure can be found in [22]). Based on the relative entropy measure described in Equation (Equation 2), we define our index for comparing food basket qualities as:(3)$$FBAI(W^{e}\left|\right|W^{r})=1-\exp[-D(W^{e}\left|\right|W^{r})].$$

The transformation defined in Equation (Equation 3) is a normalization, so 0≥FBAI≤1. Similar transformations of the KLD function have been proposed in other contexts [34]. An FBAI≈0 indicates that $W^{e}$ and $W^{n}$ are approximately the same.

As an element of getting us closer to a robust approach in inequality research, the choice of a (decomposable) relative entropy measure is guided by its proximity to the literature on information theory [35], characterizing the relative (Shannon) entropy as a measure of surprise. Entropy is a measure of concentration of probabilities. Low entropy distributions are more concentrated, and hence more informative, than high entropy distributions. Thus, our proposal is to measure food basket quality as the additional amount of information contained in the empirical food basket that is not included in the normative: a measure of how surprised the researcher is to see a household’s food acquisitions, given the common agreement of what an adequate basket for that household would look like. It is this information discrepancy between the empirical and the normative food baskets that we propose as a metric comparable across an otherwise utterly dissimilar set of distributions. Given the nature of our data, we have applied this approach at the household level, and averaged across households to assess larger population groups. The Stata syntax used to produce the results is included as Appendix A.

### 2.3. Data Analysis

In analysing the structure of the food basket, the 247 food items were grouped into 14 common FGS for better coupling with the NFB construction [28]: (1) Grains; (2) Roots and tubers; (3) Legumes; (4) Nuts and seeds; (5) Vegetables; (6) Fruits; (7) Meat; (8) Processed meat; (9) Fish; (10) Dairy products; (11) Milk; (12) Eggs; (13) Fat and oils, and (14) Sugars (Appendix A shows the food items included in each food group). The NFB leaves out seasoning herbs, spices and condiments, and other items considered obstacles for healthy diets (junk food and low-nutrient foods). Subsequently, the group structure of food acquisitions is expressed in relative terms as weight shares (i.e., as the percentage of total food kilograms acquired of each of 14 food groups). Focusing on weight shares allows us to (1) leave some room for individual (household) variability in the composition of the basket in terms of specific foods in each food group; and (2) improve the comparability across population groups with different spending levels on Food Away From Home (FAFH).

Using Equation (Equation 3), we estimated the FBAI for the 14 food groups in order to make comparisons between the normative and the empirical distributions. Furthermore, we estimated the FBAI by income deciles and presented contour plots for better visualisation of the findings when using households as units of analysis.

## 3. Results

A description of the characteristics of the 2129 households included in this study are presented in Table 1.

### 3.1. Food Basket Adherence Index Distribution

Overall, as per the NFB, our results suggest that Mexico City’s households acquire too little or few fruits, milk, vegetables, dairy products, fish, legumes, roots and tubers, sugars, and nuts and seeds, and too many meats, grains, eggs, processed meat, and fats and oils (see Table 2). In a wide distribution of scores across households (5th percentile = 0.19; 95th percentile = 0.78; median = 0.42, SD = 0.18), the average FBAI was 0.44, far above the minimum score of 0 which would indicate full adherence to the NFB.

Additionally, there are significant differences and patterns in dietary components as the FBAI worsens. Figure 1 shows the relative excess or defect the average household exhibits by FGS over deciles of the FBAI. In every panel, we see a dashed reference line indicating the normative level associated with every FGS. At the top, we can see, for example, that even those households best ranked (on the far left) acquire grains in excess (reference line always at the bottom of the panel), and this only grows worse as the FBAI increases, to more than double the norm at the far right. This behaviour is mimicked in the cases of eggs and processed meat. In contrast, with an opposite behaviour, we find most notably vegetables and fruits, falling almost absolutely short of the norm at the far right. As this behaviour suggests significant associations with socioeconomic factors [36,37,38], differences across income levels were examined.

### 3.2. Income

Figure 2 somewhat corroborates the expected pattern of fruit and grains acquisitions as we move from lower (left) to upper (right) income deciles—poor households are much more likely to acquire no fruit, and large amounts of tortillas—while it also reveals more complex dynamics in other FGS.

At the far right we can see that the richest households do exhibit better access to fruit, fish, and dairy products, while, at the same time, they comply the worst with legumes, vegetables and tubers. In addition, regarding the excess acquisitions of meats and processed meats, differences across income levels are rather small, with all deciles falling well short of the norm for milk (without distinguishing between whole and skimmed).

This worsening of food basket adherence at higher deciles of the income distribution is best depicted in Figure 3, where we can see the best adherence to the NFB around the 7th decile.

### 3.3. Household Distribution

Figure 4 allows us to better appreciate the distribution of Mexico City’s households across both our FBAI and income distributions. In the upper left corner of the heatmap, we can see in red what are most likely to be households living in food poverty: those whose income does not suffice to afford adequate food. It is important to note that this heat pocket reaches well into the 3rd income decile. Near the lower right part of the heatmap, we can see another heat pocket where households with the best food basket adherence are concentrated around the 7th and 8th income deciles. The upper middle bluish part of the heatmap is what one would expect of income brackets with adequate food access, starting in the 4th income decile. However, the two upper income deciles might be more difficult to interpret, with some red areas at the top and blue in the lower half.

### 3.4. Food Away from Home

The pattern shown in Figure 4 might have to do with the fact that households in the upper income deciles tend to spend relatively more on Food Away From Home (FAFH), weakening the relationship between home food environment and adherence to the NFB. Figure 5 allows us to dig a little deeper into this possibility. Here we can see how, indeed, we have estimated the worst home food environments in those households where FAFH expenditure is more than double Food at Home (FAH) expenditure. However, it is important to note that this does not necessarily mean that these households observe a better adherence than what their home food environment would suggest; quite the opposite, it is probable that, especially for the lower income deciles, food consumed away from home (most likely on the street) is of a rather poor quality [39,40].

The fact that hardly any of the households whose FAFH expenditure more than doubles their expenditure on FAH can be found among those we estimate to have the best home food environments (notice how all the red areas are in the upper part of Figure 5) makes us consider the possibility that our estimates are most reliable for households which spend no more than half of their of food expenditure away from home (blue and green zones in Figure 5).

### 3.5. Household Structure

Another factor that could potentially weaken the relationship between home food environment and adherence to NFB is household structure. Economies of scale exist within households, perhaps especially in food preparation; that is, it does not cost a family of four twice as much as a family of two to procure adequate food for meeting the dietary needs of its members [41,42].

Figure 6 shows an interesting picture, as it provides useful information regarding those households in greater need. From it, together with Figure 4, we can see that households constituting up to three members in the first income decile are having the worst time providing themselves with adequate food (upper left corner), while households constituting four or more persons seem to be doing better in general. It is also interesting to note that those households for which the home food environment is less informative of their food basket adherence—those with higher FAFH/FAH ratio—fall on the smaller side.

### 3.6. Food Security

Our FBAI also adds further content to the Mexican Official Food Access Deprivation Measure (OFADM) and this opens new questions. Valencia-Valero and Ortiz-Hernández [43] and Vega-Macedo et al. [37] have already shown not only that most food groups are harder to find (lower variety) in food-insecure households, but also that they are usually replaced with energy-dense foods.

Figure 7 shows the distribution of those households with Food Access Deprivation according to the Mexican Official Poverty Measure (MOPM) by income and our FBAI deciles. As expected, the greatest concentration of deprived households can be found in the upper left corner, but a couple of hotspots on the lower right corner come as a surprise. The apparent misalignment between the OFADM and households’ adherence to the city’s NFB deserves our attention. Perhaps assessing the home food environment will make it possible to shed new light on the puzzling “anomalies” in the Mexican Food Security Scale [44,45]: unexpected combinations of (high/low) socioeconomic stratum and (severe/minimal) food insecurity.

## 4. Discussion

This paper presents a new entropy-based approach for producing comparable and decomposable (across FGS) diet adherence indices drawn from different diet or food standards. We provide an empirical demonstration of the approach using Mexico City’s NFB as a diet standard. The results show that, relative to the suggested standard, households fall short in the acquisition of fruits, vegetables, dairy products, fish, legumes, roots and tubers, sugars, and nuts and seeds, but are over-acquiring meat, grains, eggs, processed meat, and fats and oils. Although we used data on expenditure at the household level, our results are in line with evidence examining adherence to diet standards in Mexico using individual intake, where findings also suggest under-consumption of fruits and vegetables, legumes, dairy products and seafood [46,47].

When examining the possible factors affecting the discrepancy between food acquisition and the NFB, we found that income plays an important role. Our estimates strongly suggest that lower-income households are less likely to procure adequate food, downgrading from the 7th decile of the income distribution. These findings further establish the link between income and the types of foods purchased or otherwise acquired by households [36,37,43]. We take this as evidence that the divergences reported do not reflect mere differences in personal tastes, but a real disadvantage to affected people; the dietary expression of social exclusion.

The example based on real data from Mexico City shows the scope of implementation of the proposed approach to widely available expenditure data. However, our approach is not restricted to such data, and future research should explore its implementation using intake data.

Our results are also important because they add significant detail to our understanding of the home food environments of food-insecure households. Our results show that there are significant and measurable differences by food security status in the food baskets acquired, with potential explanatory power for long-perceived “anomalies” in the Mexican Food Security Scale (misalignments with income, variability measures and frequency items) [44].

The application of our expenditure-based approach has several strengths. First, it relies mostly on data sources that are readily available in most low-and middle-income countries (LMIC), naturally dovetailing with poverty measures, and has potential as a means of evaluating large social development programs. Second, it takes advantage of the strengths of expenditure data sets, namely, that they rely less on memory and are less prone to social desirability biases. Third, being based on the food group and staples level, the results are not only more interpretable compared with nutrients, but also easier to translate into recommendations and intervention targets [48].

It is important to acknowledge the difference between food acquisition and intake (we looked at what people procure for their household, not what they actually eat), and even though there are good reasons to expect important correlations, our results warrant some caveats. Firstly, our approach does not directly compare with diet indices in that (i) it is based on food acquisitions as opposed to intake, and (ii) it is measured at the household as opposed to the individual level. This type of analysis assumes that food purchases are distributed according to need across all household members. However, this is not always the case, especially in food-insecure households where adults may try to protect children from food insecurity [49].

While expenditure data sets constitute a unique and valuable source of data regarding the home food environment, there are several limitations inherent in their data collection processes. Expenditure surveys usually do not include information on how households prepare meals or snacks with what is acquired, and hardly ever register food inventories that households currently have in the surveying week, or how much of the acquired food regularly goes to waste. Third, it is hardly possible to infer the quality of FAFH purchases. In this respect, some degree of caution is warranted when interpreting home food environments as an indicator of diet quality, particularly when FAFH expenditure is sizeable relative to that of food at home.

This study provides a novel approach to analyse adherence to diet standards, and this has a number of implications for future research. For instance, the entropy-based DAI could be used to evaluate the adherence to diet quality indices in Mexico, namely the Mexican Diet Quality Index [36,50], but this method can also be widely used with other diet quality indices (e.g., Healthy Diet Index [51], Alternate Healthy Eating Index [52]), diet scores (Dietary Approaches to Stop Hypertension Score [52]), and countries’ or regions’ specific dietary guidelines and using expenditure or dietary intake at individual level.

We see the proposed entropy-based DAI as having a wide range of applications, perhaps most importantly in assessing the growing concern worldwide of food security. Indeed, assessing home food environments may add further content to measures of food security in vulnerable groups. As people experiencing food insecurity usually face difficulties in acquiring food and are forced to compromise on the quality and/or quantity of the food they procure, the entropy-based DAI may prove to be a simple tool to gauge the extent of food budget restraints, even outlining the sociodemographic groups most at risk of experiencing food insecurity. In this way, the entropy-based DAI can help public officials look with higher resolution into a broader range of food insecurity levels, informing the design of appropriate interventions for specific population groups in situations of widespread economic turbulence.

On the nutritional side, the promotion of healthy diets still needs to be provided to the population. The findings presented here can inform decisions about which food groups need to be promoted in Mexico City and to which populations. For instance, the consumption of fruits and vegetables still needs to be widely promoted to all sectors of the population, even though different campaigns to increase intake of fruit and vegetables have been carried out in previous years [47]. Less is known about what is impeding household expenditure on fruits and vegetables in Mexico City. Potential factors may be the cost and access that households have to these food items, or that people are consuming these foods outside the home environment. In Mexico it is common to buy fresh fruits and vegetables from street vendors. Our findings also suggest excessive consumption of meats and processed meats in all income deciles. Although controversial, the over-consumption of red and processed meat has been considered detrimental to health, increasing the risks of diabetes, cardiovascular disease and colorectal cancer [53]. Education to this end can inform the population to reduce red and processed meat consumption and increase other dietary sources of protein, such as legumes and fish, two food groups also showing under-consumption.

It is important to keep in mind some obvious limitations in the construction of diet adherence indices. First of all, they are all predicated on the existence of shared diet norms by the target population. Our approach requires, additionally, for these norms to be meaningfully translatable as consumption shares for each food group. Second, all diet adherence indices embody a particular evaluation and metric of the “costs” of deviations from the norm, including presumed measures of nutritional substitutability that may or may not be appropriate. Our proposal is no different in this sense. The measurement of diet adherence requires a more careful discussion of this and related issues.

A less obvious limitation of our approach is that, as an application of information-theoretic concepts and instruments to the analysis of socioeconomic systems, our proposal is rather modest in defining Kullback–Leibler (or Bergman divergences more broadly) for pairs of frequency distributions of food mass across food groups for any given aggregation of individuals. Admittedly, our approach has little to say regarding the functioning of the underlying systems generating the distributions in question. dos Santos and Wiener [54] make a compelling argument regarding the careful consideration researchers must give to the construction of the domains over which they are defining distributions. Our approach effectively assumes that the FGS are the domain supporting distributions of “units of mass”. However, in the analysis of socioeconomic systems, using people as (indistinguishable) units of analysis allows for deeper analytical insights regarding their probabilistic content. Future research should pursue this relatively unexplored line of study.

## 5. Conclusions

Our entropy-based method to produce Diet Adherence Indices is a promising measurement tool, but considerable research is needed to continue to explore how it can be most useful. It remains a pending task to assess the time stability of our measures, for which longitudinal data are crucial. In addition, the development of NFBs for different dietary communities (gastronomic regions in Mexico) will surely throw some light on the comparability of our measures across different diet cultures. Further research is also needed to bridge the knowledge gap between home food environment and diet behaviours, perhaps somewhat immediately by way of examining covariations with health outcomes. It would also be desirable to conduct a joint study using individual food intake data. Combining the results of both of these kinds of analysis could help us to achieve deeper insights into the links between home food environments and the general population’s dietary habits. Ultimately, it is our hope that the work presented here will open new lines of communication between prominent research branches sharing the same interests: food poverty and diet quality.

## Figures and Tables

**Figure 1 entropy-25-01258-f001:**
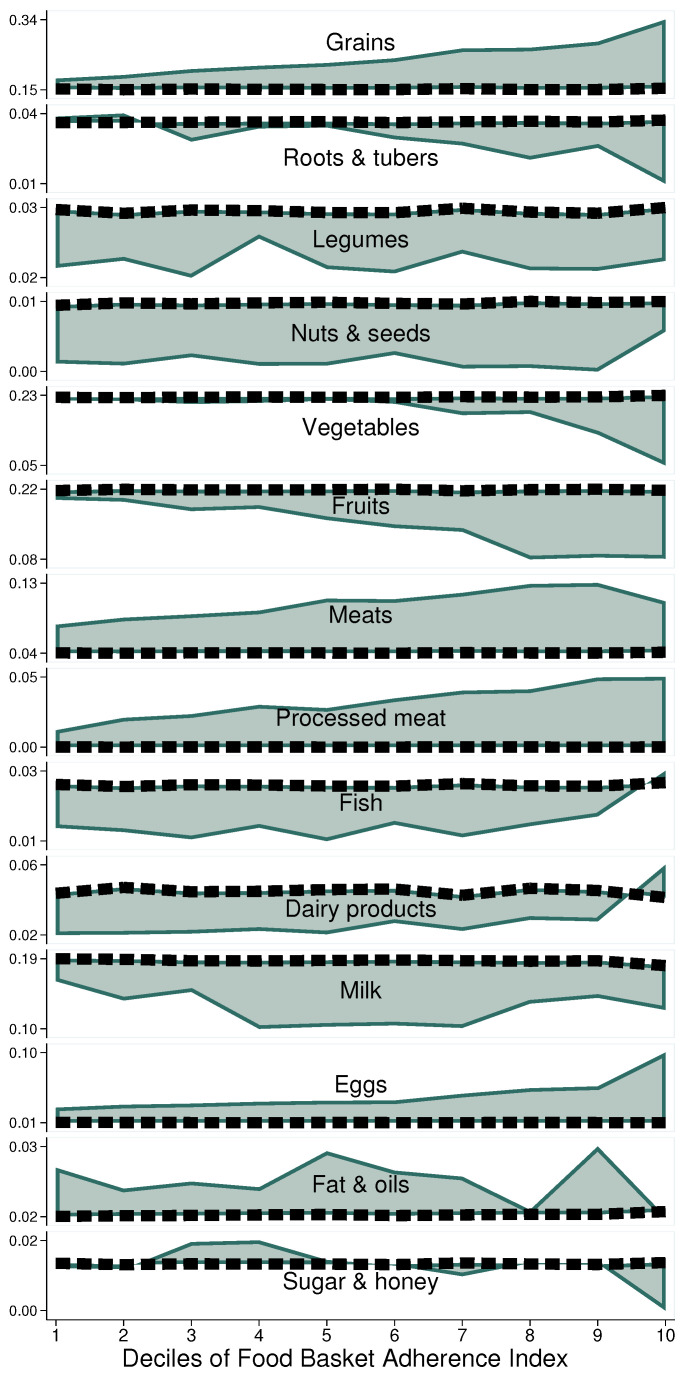
Adherence to NFB by food group over diet adherence level, Mexico City, 2018. Source: Authors’ own elaboration using data from ENIGH 2018 [32] and Ávila Curiel [28].

**Figure 2 entropy-25-01258-f002:**
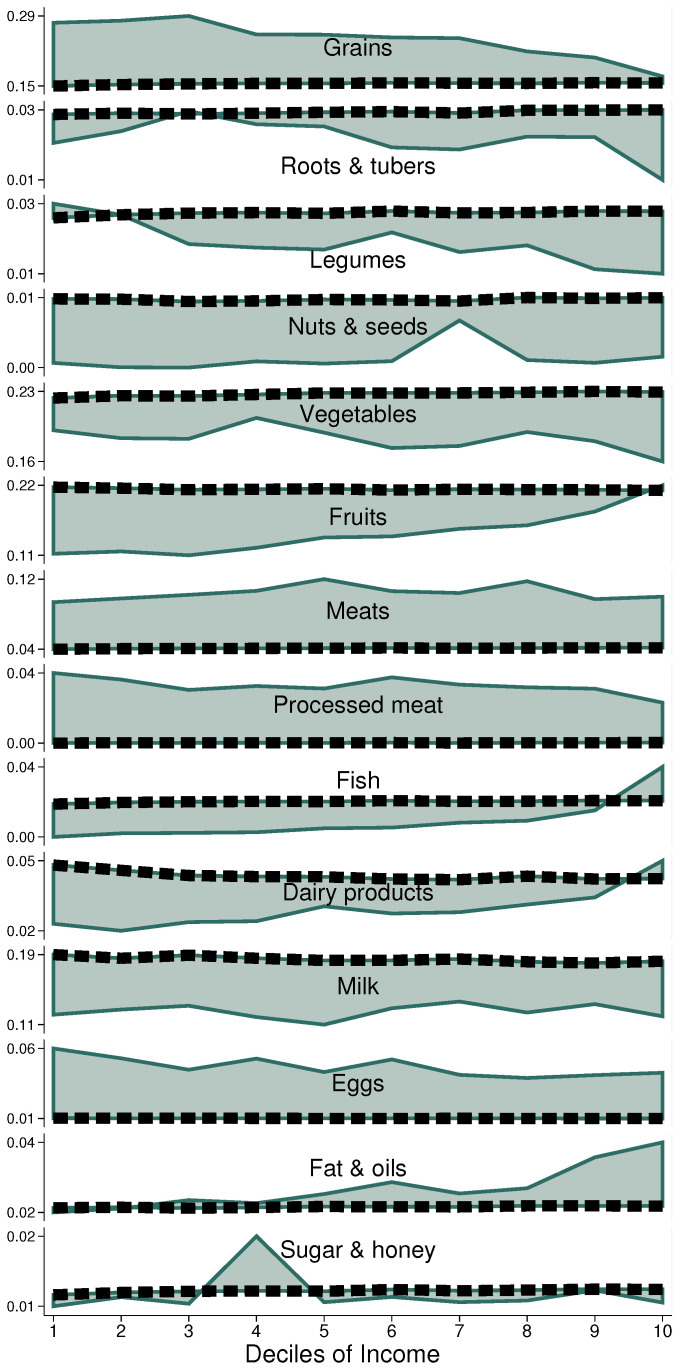
Adherence to NFB by food group over income level, Mexico City, 2018. Source: Authors’ own elaboration using data from ENIGH 2018 [32] and Ávila Curiel [28].

**Figure 3 entropy-25-01258-f003:**
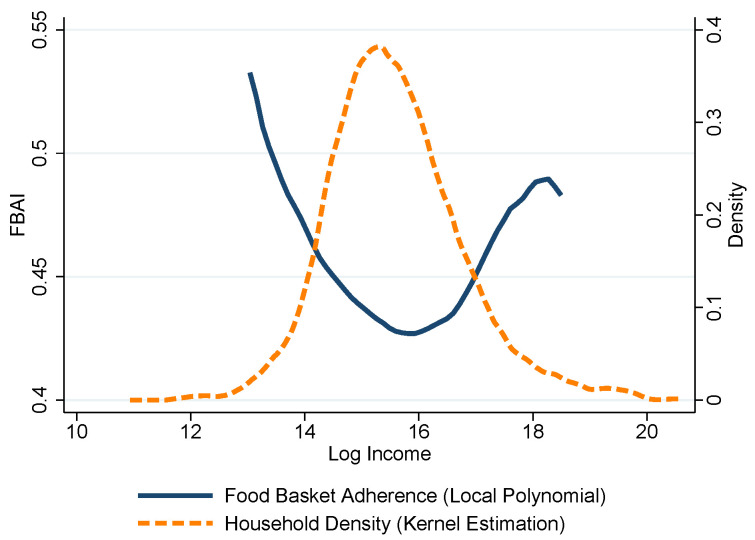
Household food acquisition adherence and income, Mexico City, 2018. Source: Authors’ own elaboration using data from ENIGH 2018 [32] and Ávila Curiel [28].

**Figure 4 entropy-25-01258-f004:**
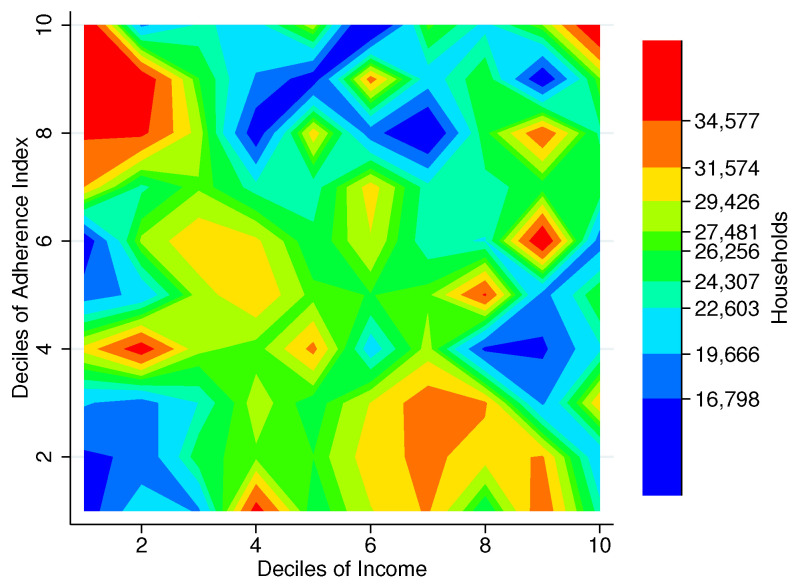
Density of households by deciles of Food Basket Adherence Index and income, Mexico City, 2018. Note: The figure shows a contour plot (sometimes called level plot) of a three-dimensional surface (Income deciles, FBAI deciles, # Households). It graphs income deciles on the *x*-axis, FBAI deciles on the *y*-axis, and the number of households at the intersection of each combination of these deciles as color-filled contours increasing from blue (<16.8 K) to red (>34.6 K). Source: Authors’ own elaboration using data from ENIGH 2018 [32] and Ávila Curiel [28].

**Figure 5 entropy-25-01258-f005:**
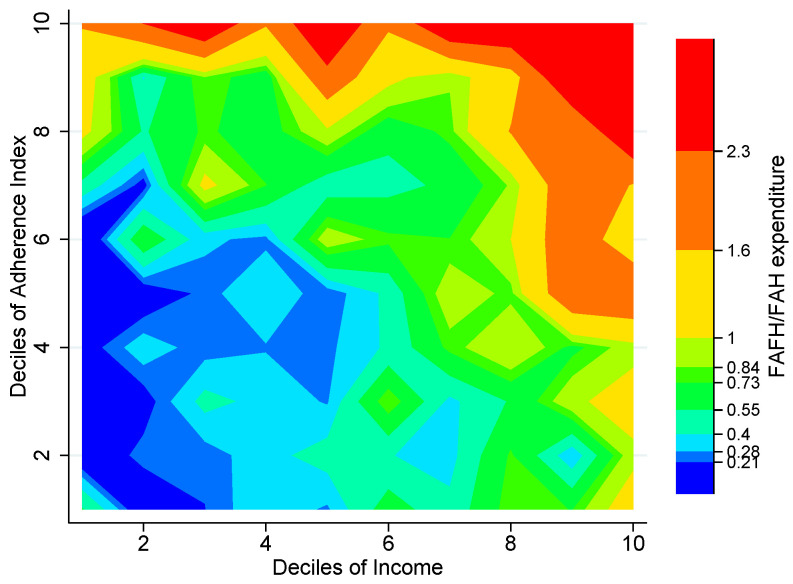
Relative food-away-from-home expenditure by deciles of FBAI and income, Mexico City, 2018. Note: The figure shows a contour plot (sometimes called level plot) of a three-dimensional surface (Income deciles, FBAI deciles, FAFH/FAH expenditure). It graphs income deciles on the *x*-axis, FBAI deciles on the *y*-axis, and the ratio of Food Away from Home (FAFH) expenditure to that of Food at Home (FAH) at the intersection of each combination of these deciles as color-filled contours increasing from blue (<0.21) to red (>2.3). Source: Authors’ own elaboration using data from ENIGH 2018 [32] and Ávila Curiel [28].

**Figure 6 entropy-25-01258-f006:**
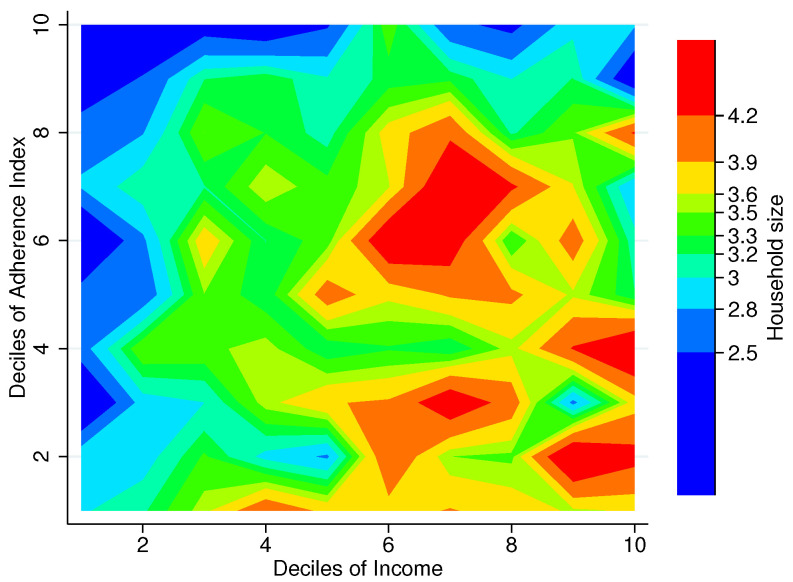
Household size by deciles of Adherence Index and income, Mexico City, 2018. Note: The figure shows a contour plot (sometimes called level plot) of a three-dimensional surface (Income deciles, FBAI deciles, Household size). It graphs income deciles on the *x*-axis, FBAI deciles on the *y*-axis, and the household size at the intersection of each combination of these deciles as color-filled contours increasing from blue (<2.5) to red (>4.2). Source: Authors’ own elaboration using data from ENIGH 2018 [32] and Ávila Curiel [28].

**Figure 7 entropy-25-01258-f007:**
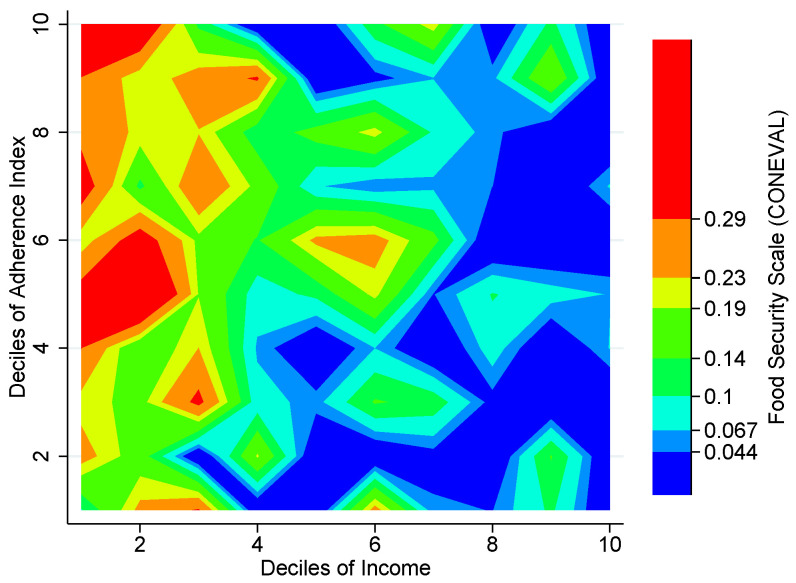
Mexican Food Security Scale (Food Access Deprivation) by deciles of Adherence Index and income, Mexico City, 2018. Note: The figure shows a contour plot (sometimes called level plot) of a three-dimensional surface (Income deciles, FBAI deciles, Food Access Deprived Households). It graphs income deciles on the x-axis, FBAI deciles on the y-axis, and the proportion of Food Access Deprived households at the intersection of each combination of these deciles as color-filled contours increasing from blue (<2.5) to red (>4.2). Source: Authors’ own elaboration using data from ENIGH 2018 [32], Ávila Curiel [28] and CONEVAL.

**Table 1 entropy-25-01258-t001:** Main characteristics of the sample.

Household Characteristics	n/%	Standard Error
Total households in sample	2129	n/a
Total gross (sampling weights) households	2.7 million	n/a
Mean household size	3.4 members	(0.001)
Mean household per capita monthly income	6750 MXN	(4.06)
	(305 USD)	
Proportion male headed households	68%	(0.005)
Proportion urban households	80%	(0.004)
Proportion households with food insecurity	15%	(0.004)

**Table 2 entropy-25-01258-t002:** Household-average normative and empirical shares (%) by food groups and staples, Mexico City, 2018.

	Normative	Empirical	KLD
FGS	$\left(w_{n}^{r}\right)$	$\left(w_{n}^{e}\right)$	$\left(D_{n}\right)$
Fruits	21.59	14.42	0.0913
Milk	18.30	12.37	0.0903
Vegetables	22.51	18.18	0.0537
Dairy products	4.55	2.85	0.0352
Fish	2.34	1.30	0.0267
Legumes	2.96	1.95	0.0254
Roots & tubers	3.27	2.55	0.0238
Sugar	1.15	1.11	0.0196
Nuts & seeds	0.49	0.07	0.0062
Meats	3.96	10.59	0.0756
Grains	15.00	23.98	0.0672
Eggs	1.13	4.61	0.0633
Processed meat	0.34	3.14	0.0630
Fat & oils	2.43	2.88	0.0244
FBAI			0.4463

## Data Availability

Publicly available datasets were analyzed in this study. This data can be found here: http://en.www.inegi.org.mx/programas/enigh/nc/2018/ (accessed on 16 February 2023).

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
