# Peer review of "An Entropy-Based Approach to Measuring Diet Adherence"

_entropy, 2023, doi:10.3390/e25091258_

Round 1

Reviewer 1 Report

The authors propose an approach  to calculate and sum up the disperancies between the actual food acquisition or intake and any standard or norm. Subsequently, they apply their approach to data collected from the Mexican National Survey of Household Income and Expenditure.

There are major issues with the manuscript such as a related works section missing. Moreover there is neither a formal or computational evaluation of proposed metric. The metric is proposed and subsequently applied without being evaluated for its soundness.

Reviewer 2 Report

First of all, I would like to thank you for reviewing this research. 

Before proceeding to the publication of the research, I think it is necessary to mention a couple of things. 

The introduction should better contextualise the type of diet followed in Mexico, as nothing is shown. In addition, together with the research objectives, the initial starting hypotheses should be established. 

Regarding the methodology, it should be readapted in the following parts: Methodology of the study, Ethical principles, Procedure and Data analysis. Also, specify and justify why these sites were chosen rather than others. 

Add the limitations and future perspectives of this study.  

Round 2

Reviewer 2 Report

The article has been considerably improved, so I am giving the go-ahead for publication.